# The Effect of Breastfeeding in the First Hour and Rooming-In of Low-Income, Multi-Ethnic Mothers on In-Hospital, One and Three Month High Breastfeeding Intensity

**DOI:** 10.3390/children10020313

**Published:** 2023-02-07

**Authors:** Lawrence Noble, Ivan L. Hand, Anita Noble

**Affiliations:** 1Department of Pediatrics, Icahn School of Medicine at Mount Sinai, Mount Sinai Hospital, New York, NY 10029, USA; 2New York City Health & Hospitals Elmhurst, New York, NY 11373, USA; 3Department of Pediatrics, New York City Health & Hospitals/Kings County Hospital, Brooklyn, NY 11203, USA; 4SUNY-Downstate College of Medicine, Brooklyn, NY 11203, USA; 5Department of Nursing, Henrietta Szold Hadassah/Hebrew University, Jerusalem 91120, Israel

**Keywords:** breastfeeding, baby friendly, rooming-in, newborn, skin-to-skin

## Abstract

Despite the known benefits of exclusive breastfeeding, the value of Baby-Friendly Hospital Interventions in increasing breastfeeding rates has been challenged, particularly the interventions of breastfeeding in the first hour of life and rooming-in. This study aimed to measure the association of breastfeeding in the first hour of life and rooming-in with high breastfeeding intensity of low-income, multi-ethnic mothers intending to breastfeed. A prospective, longitudinal cohort study was performed on 149 postpartum mothers who intended to breastfeed their infants. Structured interviews were performed at birth and one and three months. Breastfeeding intensity was defined as the percentage of all feedings that were breast milk, and high breastfeeding intensity was defined as a breastfeeding intensity >80%. The data were analyzed by chi-square, t-test, binary logistic regression analysis, and multivariate logistic regression analysis. Breastfeeding in the first hour was associated with increased high breastfeeding intensity in the hospital (AOR = 11.6, 95% CI = 4.7–28.6) and at one month (AOR = 3.6, 95% CI = 1.6–7.7), but not at three months. Rooming-in was associated with increased high breastfeeding intensity in the hospital (AOR 9.3, 95% CI = 3.6–23.7) and at one month (AOR = 2.4 (1.1–5.3) and three months (AOR 2.7, 95% CI 1.2–6.3). Breastfeeding in the first hour and rooming-in are associated with increasing breastfeeding and should be incorporated into practice.

## 1. Introduction

The American Academy of Pediatrics’ (AAP) 2022 Breastfeeding Policy Statement has recommended that infants exclusively breastfeed for six months, followed by continued breastfeeding, while starting complementary feeds for two years or beyond, as desired by mother and child [1]. There is robust evidence that the medical and neurodevelopmental advantages of breastfeeding, both short-term and long-term, make breastfeeding a public health imperative [1]. Research has revealed that children who were breastfed as infants have decreased otitis media, lower respiratory illnesses, acute diarrheal disease, sudden infant death syndrome, neonatal and infant mortality, atopic dermatitis, inflammatory bowel disease, diabetes mellitus, childhood leukemia, asthma, and obesity. Many of these infant benefits are dependent on longer durations of breastfeeding. In addition, longer durations of breastfeeding could be more critical for maternal outcomes. Research has found that breastfeeding for longer than 12 months is associated with decreasing maternal breast cancer, ovarian cancer, hypertension, and diabetes mellitus [1]. Although most U.S. infants initiate breastfeeding (83%), only 45% are exclusively breastfeeding at three months, and only 36% are still breastfeeding at one year [2]. In addition, there are critical sociodemographic and cultural differences in the breastfeeding rates. Mothers eligible for the Special Supplemental Nutrition Program for Women, Infants, and Children (WIC), those with a high school education or less, and younger mothers, under 20 years of age, all have lower breastfeeding rates [2]. Therefore, specific hospital practices that can increase breastfeeding in these populations need to be identified.

The recent Centers for Disease Control (CDC) data has revealed that higher Maternity Practices in Infant Nutrition and Care (mPINC) scores, which reflect increased Baby Friendly Hospital Initiative (BFHI) hospital practices, have been associated with increased exclusive breastfeeding at hospital discharge [3] and at eight weeks [4]. The Agency for Healthcare Research and Quality (AHRQ) recently reviewed 40 studies and concluded that BFHI was associated with higher rates of breastfeeding initiation and duration [5]. In addition, an intervention to improve hospital policies, consistent with the Ten Steps, in the southern U.S. increased the breastfeeding initiation of African-American infants from 46% to 63% (*p* < 0.05) and exclusivity from 19% to 31% (*p* < 0.05), respectively. It also decreased breastfeeding disparities between African-American and White infants by 9.6% [6]. However, studies on the effectiveness of BFHI from Australia [7], DR Congo [8], and a recent U.S. population-based study [9] did not show increased breastfeeding rates. A 2016 review of 25 studies on the effectiveness of BFHI reported that, due to the limitations of the studies, there needs to be more evidence to conclusively conclude that BFHI improves breastfeeding initiation, duration, or exclusivity [10].

Recent articles have challenged specific BFHI interventions as not being evidenced-based and, in addition, are associated with risks, particularly breastfeeding in the first hour of life and rooming-in [11]. Breastfeeding in the first hour and rooming in have been accused of being risky to the infant, as it could increase the risk of infant death from sudden unexpected postnatal collapse [11]. A 2016 Cochrane review of rooming-in was able to review only one study and reported that there is not enough evidence to recommend or not recommend rooming-in based on the current studies [12]. 

There is a shortage of studies evaluating the specific BFHI interventions of breastfeeding in the first hour of life and rooming-in on subsequent breastfeeding rates. In addition, measuring the effect of breastfeeding in the first hour of life and rooming-in on high breastfeeding intensity (whether 80% of all feeds are breast milk feeds) has not been consistently evaluated. The outcome measure of high breastfeeding intensity may better evaluate the slight differences in breastfeeding outcomes, particularly in populations, such as the study population with a history of mixed feeding. The purpose of the current study was to evaluate the impact of breastfeeding in the first hour of life and rooming-in in the hospital on high breastfeeding intensity for low-income, inner-city mothers who expressed an intent to breastfeed in the immediate postpartum period.

## 2. Methods 

A prospective, analytical cohort study was conducted on postpartum mothers who intended to breastfeed their infants. The eligibility criteria included: maternal age of 18 years or older, intent to breastfeed, stable residence in the previous two years, a working phone number, and the birth of a singleton, healthy-term infant. The exclusion criteria included: any medical/obstetrical complications for which breastfeeding is contra-indicated (e.g., HIV, HTLV-1, HTLV-2) or participants not fluent in English or Spanish. The participants were recruited and screened for eligibility using procedures that conform to the HIPAA requirements and institutional IRB practices, including informed consent. The Icahn School of Medicine at Mount Sinai Institutional Review Board approved the study. During the four month enrollment period, participants were recruited from the postpartum floor, five days per week, at Elmhurst Hospital, which serves a multicultural patient population in New York City. A Spanish-English bilingual research coordinator was qualified in a standard script and administration protocol, interviewing techniques, and data collection procedures. The baseline interview was conducted face-to-face after written informed consent was obtained.

The study instruments included a baseline questionnaire and follow-up questionnaires at 1 and 3 months of age, adapted from the Infant Feeding Practices Survey II [13]. The baseline questionnaire was completed as part of a structured interview with the participants during their postpartum hospitalization. It included demographic, socioeconomic, medical, and breastfeeding information. The questionnaires at 1 and 3 months of age were completed by phone interviews and assessed breastfeeding and other feedings. All of the tools and consent forms not previously published in Spanish were translated using Brislin’s method of translation [14]. Any discrepancies in meaning were corrected by consensus on meaning and intent by bilingual healthcare professionals with expertise in lactation. The baseline interview included an assessment of the postpartum breastfeeding interventions, breastfeeding in the 1st hour, and rooming-in. These questions were repeated at the one-month interview, as the initial interview could have taken place early in the hospital stay and may not have included what transpired for the rest of the hospital stay. If there was a discrepancy between the responses in the baseline and one month interviews, the one month response was used. Breastfeeding in the 1st hour of life was assessed by the question, “When did you first put the baby to breast?” Rooming-in was optional and was assessed by asking the question, “How often does the baby room-in with you?” followed by the choices: “all of the time”, “some of the time”, or “not at all”. For this study rooming-in was defined as “all of the time.”

High breastfeeding intensity was compared at birth, one month, and three months. Birth was defined as the entire postpartum hospital stay. Breastfeeding intensity was defined as the percentage of all feedings that were breast milk and high breastfeeding intensity as a breastfeeding intensity >80%. If both breast milk and formula were offered in the same feeding, the feeding was scored as 0.5 breast milk and 0.5 formula. The postpartum phone interviews at one and three months assessed infant feeding in the previous seven days. The 1 month interview was performed at 30 ± 7 days, and the 3 month interview at 90 ± 21 days.

A power analysis was performed for an effect size of 0.5, alpha 0.5, power of 80%, and a 20% attrition rate. The data were analyzed by chi-square, t-test, and binary logistic regression analysis with SPSS 27. Maternal ethnicity, foreign-born, age, immigration after 18 years, limited English skills, receiving WIC, marital status, overweight or obesity, work or school intention in the first year, gravida, breastfed in the past, and delivery type were collected in the baseline interview and included as confounding variables in the adjusted regression analysis. Maternal age was defined as age < 20 vs. 20 or above, marital status as legally married at the time of delivery, overweight/obesity as BMI 25 or above, gravida as previous live birth, and delivery type as NSVD vs. C-section or assisted vaginal delivery.

## 3. Results

There were 627 postpartum mothers screened for participation in the study, and 478 (76%) were excluded. Reasons for exclusion included less than two years of stay in residence (52%), lack of fluency in either English or Spanish (23%), decline to enroll (17%), or no intent to breastfeed (8%). Those who declined to participate in the study did not differ from the study regarding age, parity, or socioeconomic status. The probe questions asked by the research assistant indicated that most of those who declined to enroll did so out of concern for their non-documented immigration status. Of the 627 mothers screened, 149 participated in the study, and the data for the three months were obtained on 128 mothers (86%).

The study population was 67% Hispanic, of which 62% self-identified as Mexican or South American. The rest of the sample identified as Asian/Indian (17%), Caribbean-American (5%), African/African-American (5%), and Caucasian (7%). Seventy-one percent of the participants were receiving or qualified to receive WIC benefits. Eighty-six percent of the population was born outside the United States, and 69% immigrated after the age of 18. The mothers had lived in the United States for a mean of 11 years, with a range of between 2 and 42 years. Sixty percent did not speak English at home, with friends, nor read or think in English. Sixty-one percent of participants had earned a high school diploma, and 19% had a college degree.

Breastfeeding increased from 84% at birth to 92% at one month of age and then decreased to 70% at three months. Fifty-four percent (80/149) of mothers breastfed in the first hour of life, and 66% (99/149) roomed-in with their infant. The maternal characteristics associated with breastfeeding in the first hour and rooming-in are listed in Table 1. Of note, mothers who delivered vaginally and Hispanic mothers were more likely to breastfeed in the first hour, and foreign-born mothers were more likely to room-in (Table 1).

The unadjusted odds ratio (OR) of breastfeeding in the first hour and high breastfeeding intensity is shown in Table 2. Breastfeeding in the first hour of life was associated with increased high breastfeeding intensity in the hospital (70% vs. 20%, OR = 3.2, 95% CI = 1.6–6.2), one month (56% vs. 29%, OR = 3.2, 95% CI = 1.4–6.8), and three months (41% vs. 25%, OR = 2.2, 95% CI = 1.1–4.4. Rooming-in was associated with increased high breastfeeding intensity at birth (62% vs. 18%, OR 7.3, 95% CI = 3.2–16.7), one month (52% vs. 28%, OR = 2.7, 95% CI = 1.3–5.7), and three months (40% vs. 20%, OR = 2.7, 95% CI = 1.2–6.0). There were no differences between the Hispanic and non-Hispanic mothers in terms of high breastfeeding intensity in the hospital, one month and three months.

The adjusted odds ratio (AOR), after adjusting the analysis for maternal ethnicity, foreign-born, age, immigration after 18 years, limited English skills, overweight or obesity, receiving WIC, marital status, work or school intention in the first year, gravida, breastfed in the past, and delivery type, is shown in Table 3. Both natality and having breastfed in the past showed a borderline difference in high breastfeeding intensity in the hospital (*p* = 0.062 and *p* = 0.057) and no difference at one and three months. As the results were similar, we only included breastfeeding in the past in the analysis. The multivariate logistic regression analysis revealed that breastfeeding in the first hour was associated with increased high breastfeeding intensity in the hospital (AOR = 11.6, 95% CI = 4.7–28.6) and at one month (AOR = 3.6, 95% CI = 1.6–7.7), but not at three months. Rooming-in was associated with increased high breastfeeding intensity in the hospital (AOR 9.3, 95% CI = 3.7–23.7) and at one month (AOR = 2.4 (1.1–5.3), and three months (AOR 2.7, 95% CI 1.2–6.3).

## 4. Discussion

The aim of this study was to evaluate the association of breastfeeding in the first hour of life and rooming-in with high breastfeeding intensity for low-income, inner-city mothers who expressed an intent to breastfeed in the postpartum period. Due to limited hospital funding and resources in low-income communities, specific hospital interventions that significantly increase breastfeeding rates and breastfeeding intensity need to be evaluated to determine which interventions should be incorporated into practice. In our study, breastfeeding in the first hour was associated with increased high breastfeeding intensity in the hospital and at one month. In addition, rooming-in was associated with increased high breastfeeding intensity in the hospital and at one and three months. The present study’s findings increase the evidence that these Baby Friendly Hospital Interventions are essential and need to be incorporated into practice. 

The Baby Friendly Hospital Intervention of breastfeeding in the first hour begins with continuous skin-to-skin care, from birth until after the first feeding. Studies have shown that skin-to-skin care from birth stabilizes the infant’s body temperature and glucose, decreases crying, and provides cardiorespiratory stability [15]. It also has important benefits for the mother, which include faster expulsion of the placenta, less bleeding, improved breastfeeding self-efficacy, and lower maternal stress levels [16]. A Cochrane review published in 2016 concluded that skin-to-skin care increases breastfeeding, including increased duration and exclusivity [17]. Consistent with our findings, infants in India who were breastfed in the first hour had two times higher odds of being exclusively breastfed at six weeks in one study, [18] and, in another study, were more likely to exclusively breastfeed for six months and breastfeed for a longer duration [19]. Ekstrom et al. also published that infants whose first breastfeeding was early were more likely to have increased breastfeeding and exclusive breastfeeding in Sweden [20]. These studies are consistent with the present study, in which infants who were breastfed in the first hour had increased high breastfeeding intensity in the hospital and at one month. 

In addition, consistent with our findings, rooming-in in Taiwan was reported to be associated with higher exclusive breastfeeding at one and three months postpartum [21]. However, a Cochrane review published in 2016 located just one randomized controlled trial that reported on the effects of rooming-in on breastfeeding duration [12]. It reported that there is not enough evidence to recommend or not recommend rooming-in, based on the current studies. A more recent meta-analysis on rooming-in, published in 2019 by Ng et al., found no effect on the number of infants fully breastfeeding at three months, four months, and six months [22]. The present study found significant differences in high breastfeeding intensity in the hospital, at one month and three months of age, in infants who were rooming-in. These results may differ from previous studies of rooming-in that did not evaluate breastfeeding intensity, which may be a better measure to evaluate small differences in breastfeeding, particularly in populations with a history of both breast milk and formula feedings. Eighty-six percent of the study population were born outside the United States, 67% were Hispanic, and 17% Asian/Indian. Although the breastfeeding initiation rate was 84%, the exclusive breastfeeding rate in the hospital was only 23%. The conviction that formula contains “extra vitamins” and that feeding an infant both breastmilk and formula, referred to as “las dos cosas”, is healthier for a baby was prevalent not only among the Latina mothers delivering in our hospital, but also among the Indian mothers in our study population [23]. 

Breastfeeding in the first hour and rooming in have been challenged as risky for the infant [11]. There is a concern that breastfeeding in the first hour, with its requirement for continuous skin-to-skin contact immediately after birth until completion of the first feeding, and rooming-in with an exhausted or sedated mother, could increase the risk of infant death from sudden unexpected postnatal collapse. However, evidence shows that sudden unexpected postnatal collapse can be prevented through staff education and adequate supervision during skin-to-skin and rooming-in [24]. In addition, more recent evidence has shown that sudden unexpected infant death is not secondary to the initiatives associated with the Ten Steps. The data collected from the U.S. and Massachusetts revealed that the increasing number of births in Baby Friendly Hospitals between 2004–2016 increased skin-to-skin care and other BFHI interventions and, yet, was associated with a decreased prevalence of sudden unexpected infant deaths in the first seven days after birth [25]. Therefore, there is little evidence that breastfeeding in the first hour and rooming-in are causally linked to sudden unexpected postnatal collapse.

Another recent challenge to breastfeeding in the first hour and rooming-in has been COVID-19. During the initial phase of the COVID-19 pandemic, in 2020, the AAP and CDC recommended that newborns be separated from their mothers immediately after birth and through the whole hospital course, unless the mother could be proven negative for COVID-19 [26]. Newborn infants lost the opportunity for breastfeeding in the first hour, rooming-in, and direct breastfeeding. As evidence emerged that it was safe for infants to room-in with their mothers and to breastfeed [27,28,29], the policy was changed. However, the evidence reveals that the earlier COVID-19 policy was associated with decreased breastfeeding, while that protocol was enforced. A retrospective cohort study of mothers with COVID-19, conducted via an online survey performed between May and September 2020, found that infants who did not receive skin-to-skin and breastfeeding in the first hour, did not room-in, or did not directly breastfeed during the initial hospitalization had decreased exclusive breastfeeding at three months (AOR 0.38, 0.26, and 0.17, respectively). They also found that nearly 60% of mothers who were separated from their infants felt “very distressed” and that 29% who tried to breastfeed after the initial hospitalization were unable. A study of eleven hospitals in Massachusetts revealed that the separation of mother and infant was associated with decreased breastfeeding in the first month of life. In addition, a study from Los Angeles reported that mothers who delivered during the pandemic had lower rates of any breastfeeding at three and six months and fully breastfeeding at one, three, and six months compared to mothers who delivered prior to the pandemic. These findings from the pandemic are consistent with the results of our study, while also highlighting the importance of keeping mothers and infants together after birth to protect breastfeeding during future pandemics, unless there is clear evidence of harm. 

The study population was overwhelmingly immigrant and poor. Seventy-one percent of the participants were receiving or qualified to receive WIC benefits. Eighty-one percent had a high school education or less, and 45% were unmarried. Disparities in breastfeeding have been found among mothers who receive WIC, those with a high school education or less, and those who are unmarried. These disparities may be due to bias in obtaining prenatal or hospital breastfeeding support or to the lack of readily available lactation support after leaving the hospital for mothers in many inner-city areas [30]. Many inner-city families cannot find or afford private lactation consultants. In addition, peer support groups, such as La Leche, are often not easily found in inner-city or immigrant communities. Therefore, it was crucial to identify hospital practices, such as breastfeeding in the first hour and rooming-in, that were associated with increased high breastfeeding intensity in this population.

A potential recall bias in the maternal self-reporting of breastfeeding limited this study. This recall bias may have resulted in non-differential misclassification or over-report of breastfeeding. The former would bias toward the null hypothesis. A time-adjusted recall period window was used to reduce the potential for such bias. We considered a 24 h recall and a diary method as alternatives. We decided against a 24 h recall as it may not be representative of the usual feeding. We also decided against feeding logs, which require daily notes, and were thought to be unfeasible in this population. The lack of random assignment of the participants to breastfeeding in the first hour and rooming-in is also a limitation of the study. However, as the study did not intend to deny patients their right to receive the available services from the hospital that could increase breastfeeding, we chose an observational method of assessing the impact of an intervention. All of the study participants expressed an interest in breastfeeding; therefore, there was no difference in the allocation of resources. Although this study corrected for multiple confounding variables, there may have been other important variables that were not measured. 

## 5. Conclusions

Breastfeeding in the first hour and rooming-in was associated with increased high breastfeeding intensity. The present study’s findings add to the evidence that hospitals should incorporate Baby Friendly Hospital Interventions, such as breastfeeding in the first hour of life and rooming-in into practice. 

## Figures and Tables

**Table 1 children-10-00313-t001:** Maternal Characteristics for Breastfeeding in the First Hour and Rooming-In.

Maternal Characteristics	Breastfeeding in the First Hour			Rooming-In		
	Yes ^1^ N = 80	No ^1^ N = 69	*p*	Yes ^1^ N = 99	No ^1^ N = 50	*p*
Hispanic	61 (76)	41 (59)	0.027	69 (70)	33 (66)	N.S.
Foreign Born	73 (91)	58 (84)	NS	91 (92)	40 (80)	0.035
Married	45 (56)	38 (55)	NS	60 (61)	23 (46)	NS
WIC	57 (71)	49 (71)	NS	68 (69)	38 (76)	NS
HS Diploma	47 (59)	44 (64)	NS	65 (66)	26 (52)	N.S.
College Diploma	13 (16)	15 (22)	NS	22 (22)	6 (12)	N.S.
School < 1 Year	6 (8)	1 (2)	NS	7 (7)	0	N.S.
Work < 1 Year	20 (25)	25 (36)	NS	25 (25)	20 (40)	N.S.
Overweight/Obese	44 (55)	36 (52)	NS	54 (55)	26 (52)	N.S.
Was Breastfed	71 (89)	56 81)	NS	86 (87)	41 (82)	NS
BF Previous Child	52 (65)	38 (55)	NS	62 (63)	28 (56)	N.S.
Vaginal Delivery	66 (83)	39 (57)	0.001	70 (71)	35 (70)	N.S.

^1^ Data are presented by n (%).

**Table 2 children-10-00313-t002:** Unadjusted Odds Ratio of Breastfeeding in the First Hour and Rooming-In on High Breastfeeding Intensity ^1,2^.

High Breastfeeding Intensity	Breastfeeding in the First Hour				Rooming-In			
	Yes	No			Yes	No		
	% HBI ^1^	% HBI ^1^	*p*	OR (95% CI) ^3^	% HBI ^1^	% HBI ^1^	*p*	OR (95% CI) ^3^
Hospital	70%	20%	0.000	3.2 (1.6–6.2)	62%	18%	0.000	7.3 (3.2–16.7)
1 Month	56%	29%	0.001	3.2 (1.4–6.8)	52%	28%	0.006	2.7 (1.3–5.7)
3 Month	41%	25%	0.032	2.2 (1.1–4.4)	40%	20%	0.013	2.7 (1.2–6.0)

^1^ High Breastfeeding Intensity (HBI) is defined as breastfeeding intensity > 80%. Breastfeeding intensity is defined as the percentage of all feedings that were breast milk. ^2^ Univariate logistic regression analysis. ^3^ OR = Odds Ratio. 95% CI = 95% Confidence Interval.

**Table 3 children-10-00313-t003:** Adjusted ^1^ Odds of Breastfeeding in First Hour and Rooming-In on High Breastfeeding Intensity ^2,3^.

High Breastfeeding Intensity	Breastfeeding in the First Hour		Rooming-In	
	*p*	AOR (95% CI) ^4^	*p*	AOR (95% CI) ^4^
Hospital	0.000	11.6 (4.7–28.6)	0.000	9.3 (3.7–23.7)
1 Month	0.001	3.6 (1.6–7.7)	0.023	2.4 (1.1–5.3)
3 Month	NS	NS	0.022	2.7 (1.2–6.3)

^1^ Adjusted for maternal age, maternal ethnicity, foreign-born, receiving WIC, immigration after 18 years, limited English skills, marital status, gravida, overweight/obesity, work or school intention in the first year, breastfed in the past, and delivery type. ^2^ Multivariable logistic regression analysis. ^3^ High Breastfeeding Intensity is defined as breastfeeding intensity > 80%. Breastfeeding intensity is defined as the percentage of all feedings that were breast milk. ^4^ AOR = Adjusted Odds Ratio. 95% CI = 95% Confidence Interval.

## Data Availability

The data presented in this study are available on request from the corresponding author.

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
