# Peer review of "The Effect of Breastfeeding in the First Hour and Rooming-In of Low-Income, Multi-Ethnic Mothers on In-Hospital, One and Three Month High Breastfeeding Intensity"

_children, 2023, doi:10.3390/children10020313_

Round 1
Reviewer 1 Report
This study evaluated the effect of breastfeeding in the first hour and rooming-in on high breastfeeding intensity for low-income, minority mothers. The results of this study might contribute to the practice of rooming-in. However, there are still some critical issues that needs to be addressed.
1. The research gap and the significance of this topic are expected in the Introduction Section.
2. The study design should be consistent in the Methods Section and Abstract. Whether it is cohort or observational study?
3. In the Discussion Section, the authors only listed the results of previous studies, however, the reasons of the consistency or inconsistency should be further discussed.
Author Response
Reviewer 1
Thank you for your comments. Our response is written below in bold italics.
This study evaluated the effect of breastfeeding in the first hour and rooming-in on high breastfeeding intensity for low-income, minority mothers. The results of this study might contribute to the practice of rooming-in.
However, there are still some critical issues that needs to be addressed.
- The research gap and the significance of this topic are expected in the Introduction Section.
A sentence was added to the introduction section, line 35-37, on the significance of the topic and another two sentences, line 55-59, on the research gap.
- The study design should be consistent in the Methods Section and Abstract. Whether it is cohort or observational study?
The Methods Section, line 64, was changed to “cohort study.”
- In the Discussion Section, the authors only listed the results of previous studies, however, the reasons of the consistency or inconsistency should be further discussed.
This was amended in the Discussion Section (line 180-182): “These studies are consistent with the present study in which infants who breastfed in the first hour had increased high breastfeeding intensity in the hospital and at 1 month.”
Additionally, line 189-194 now states: “The present study found significant differences in high breastfeeding intensity in the hospital, at 1 month and 3 months of age in infants who were rooming-in. These results may be different than previous studies that did not evaluate breastfeeding intensity which may be a better measure to evaluate small differences in breastfeeding, especially in populations with a history of both breast milk and formula feedings..”
Reviewer 2 Report
This paper investigates an important question: how are two specific aspects of the BFHI--breastfeeding in the first hour of life and rooming in--associated with longer-term breastfeeding rates? In identifying a positive association between these two practices and high breastfeeding intensity, the authors present evidence in support of these two practices, which can have important implications for local and global policy. This is an important finding for researchers in this field, and some improvement to the methods and language used throughout the will strengthen its value to the research community.
Introduction:
- Lines 38-41: Please use the full proper names for the CDC and AHRQ before including acronyms in parentheses.
- Lines 52-5: The authors state that the goal is to evaluate the impact of these BFHI practices associated with high BF intensity among low-income, minority mothers (also maybe consider rephrasing this here and at the beginning of the discussion). However, only ethnicity is included in the final model. What are the differences between Hispanic and non-Hispanic mothers? Are there differential effects of BF within the first hour and rooming-in by ethnicity (i.e., an interaction term)? Were there final differences by natality? What are the findings when race + ethnicity are modeled?
Methods:
- Line 57: Please revise the language that the study was "performed on postpartum mothers".
- Lines 78-9: Please provide a reference for Breslin's method of translation.
- Line 79: Who were the people involved in correcting translations by consensus?
- Lines 80-5:
> What kinds of postpartum breastfeeding interventions were assessed? Why were these data not included in the final models? It seems this might be an important confounder.
> Why were BF in the first hour and rooming-in data collected at the one-month interview? What happened if there was a discrepancy between the baseline and one-month responses?
- Lines 86-9: Please clarify how High BF Intensity was calculated: what did the authors do if both breast milk and formula milk were offered in the same session? If this information was not collected, please clarify how these questions were asked.
- Lines 94-5: Please define the confounding variables: how they were collected and operationalized in the final models. How was marital status collected/classified, and why is it not included as a potential confounder?
Results:
- Lines 111-112: By "length of stay" do the authors actually mean to refer to residence in the United States? This is a highly disputed measure for acculturation (and there are validated surveys to measure this more accurately), so please justify using this as a potential proxy for acculturation.
Lines 115-6: The authors present overall breastfeeding rates over time. Was this tested as an outcome? If so, why are the results not presented? If it was not tested, why not? This seems to be an important outcome to consider.
Discussion:
Lines 176-7: In addition to potential recall bias, what are other variables that were not collected that might be important confounders, particularly in terms of breastfeeding support? Did the authors collect whether or not mothers met with a lactation consultant in the hospital or after discharge? What about perceived support for BF from family members (partners, mothers, sisters, etc.)?
Author Response
Reviewer 2
Thank you for your comments. Our responses are written below in bold italics.
This paper investigates an important question: how are two specific aspects of the BFHI--breastfeeding in the first hour of life and rooming in--associated with longer-term breastfeeding rates? In identifying a positive association between these two practices and high breastfeeding intensity, the authors present evidence in support of these two practices, which can have important implications for local and global policy. This is an important finding for researchers in this field, and some improvement to the methods and language used throughout the will strengthen its value to the research community.
Introduction:
- Lines 38-41: Please use the full proper names for the CDC and AHRQ before including acronyms in parentheses.
The full proper names for each were added when written for the first time. Please see lines 39 and 42.
- Lines 52-5: The authors state that the goal is to evaluate the impact of these BFHI practices associated with high BF intensity among low-income, minority mothers (also maybe consider rephrasing this here and at the beginning of the discussion).
The text was amended to “low-income, inner-city” mothers in the introduction (line 61) and discussion (line 163).
However, only ethnicity is included in the final model. What are the differences between Hispanic and non-Hispanic mothers?
There were no differences between Hispanic and non-Hispanic mothers in high BF intensity in the hospital, 1 month and 3 months.
Are there differential effects of BF within the first hour and rooming-in by ethnicity (i.e., an interaction term)?
Regression analysis did not show a differential effect of ethnicity on high BF intensity in the hospital, 1 month or 3 months.
Were there final differences by natality?
Both natality and breastfeeding in the past showed a borderline difference on high BF intensity in the hospital (p=.062 and p=.057) and no difference at 1 and 3 months. As the results were similar, we only included past BF in the analysis.
What are the findings when race + ethnicity are modeled?
No significant findings in the hospital, 1 month and 3 months. The study population included only 5% African American. This is noted in the results section.
Methods:
- Line 57: Please revise the language that the study was "performed on postpartum mothers".
The text was amended to “conducted,”(line 64).
- Lines 78-9: Please provide a reference for Breslin's method of translation.
This was added as reference 14.
- Line 79: Who were the people involved in correcting translations by consensus?
Healthcare professionals with expertise in English, Spanish and Lactation. Backward/forward translation was performed.
- Lines 80-5:
> What kinds of postpartum breastfeeding interventions were assessed? Why were these data not included in the final models? It seems this might be an important confounder.
This sentence was amended for clarity: The baseline interview and one-month interviews included an assessment of the post-partum breastfeeding interventions, BF in the 1st hour and rooming-in.
> Why were BF in the first hour and rooming-in data collected at the one-month interview?
The initial interviews were performed any time during the post-partum hospitalization, which could have taken place early in the hospital stay and, therefore, did not include what transpired for the rest of the hospital stay. Therefore, for rooming-in, the questions were asked again at the one-month interview in case there was a change that occurred after the time of the baseline interview.
What happened if there was a discrepancy between the baseline and one-month responses?
If there was a discrepancy we used the one-month responses as they would have reflected the full hospital course.
- Lines 86-9: Please clarify how High BF Intensity was calculated: what did the authors do if both breast milk and formula milk were offered in the same session?
As per previous studies, breastfeeding intensity was defined as the percentage of milk feedings in which the infant received breast milk, that is, (number of breast milk feedings/(breast milk feeds + formula feeds) × 100%. We defined breastfeeding intensity to be “high” if >80% of milk feedings were of breast milk.
A sentence was added (lines 96-97): If both breast milk and formula were offered in the same feeding the feeding was scored as 0.5 breast milk and 0.5 formula.
If this information was not collected, please clarify how these questions were asked.
- Lines 94-5: Please define the confounding variables: how they were collected and operationalized in the final models. How was marital status collected/classified, and why is it not included as a potential confounder?
Confounding variables were all collected in the initial interview.
Definitions: overweight/obesity is BMI <25 vs. 25 or above, gravida is yes or no previous live birth, delivery type is NSVD vs. C-section or assisted vaginal delivery, maternal age is age<25 vs 25 or above, years living in US is <10 years vs. 10 years or above, marital status is legally married at the time of delivery.
The lists of confounding variables in the 3 places where they are written (line 102-103, 145-146, and Table 3, line 154-155) wasn’t complete, and has been amended. As per the reviewer’s suggestion, the analysis now includes marital status, immigration after age 18, and limited English language skills, whereas length of stay was removed from the analysis (as per the next comment). The AOR’s in the abstract (line 24-27), results (line 149-152), and in Table 3 contain the updated analysis.
Results:
- Lines 111-112: By "length of stay" do the authors actually mean to refer to residence in the United States? This is a highly disputed measure for acculturation (and there are validated surveys to measure this more accurately), so please justify using this as a potential proxy for acculturation.
We removed “a measure of acculturation” in line 121.
We removed length of stay as a confounding variable. For acculturation, we analyzed the data collected on age of immigration (before or after age 18) and 4 ways of measuring language skills: language spoken at home language spoken with friends, language for reading and language for thinking. See line 120-122.
Lines 115-6: The authors present overall breastfeeding rates over time. Was this tested as an outcome? If so, why are the results not presented? If it was not tested, why not? This seems to be an important outcome to consider.
The study was designed to measure the outcome of high breastfeeding intensity. Although routinely breastfeeding rates and exclusive breastfeeding rates have been measured, for the past 20 years multiple breastfeeding studies have measured breastfeeding intensity and high breastfeeding intensity, which are thought to be better measures to evaluate small differences in breastfeeding, especially in populations such as ours with a history of mixed feeding.
Discussion:
Lines 176-7: In addition to potential recall bias, what are other variables that were not collected that might be important confounders, particularly in terms of breastfeeding support? Did the authors collect whether or not mothers met with a lactation consultant in the hospital or after discharge? What about perceived support for BF from family members (partners, mothers, sisters, etc.)?
Although the study wasn’t designed to measure the effect of Lactation Consultant visits, we did collect information on whether there was a visit. Lactation Consultant visits, in our study, did not increase breastfeeding. However, this is not surprising as mothers with lactation problems are more like to be those that are seen by the Lactation Consultant in a busy hospital than mothers without lactation problems. Since the study was not designed to evaluate LC visits, this information is not included in the results.
A sentence was added to the limitations section (line 206-207): “Although this study corrected for multiple confounding variables, there may be other variables that were not measured.”
Reviewer 3 Report
In physics and chemistry humaniity thought and studied what might be the ultimate building blocks of matter, and in biology, the fundamental unit of plant and animal life, the cell. With the knowledge in Medicine and biology that breastfeeding saes maternal and pediatric lives annually, the emergence of steps, 10 baby friendly not surprisingly emerged. The effectiveness of the steps, all of them, or a subset, would naturally emerge as its own area of study. The first nine Baby Friendly steps seem more valuable in the Congo based on a Melinda and Bill Gates supported study, for example. This study examines 149 post partum Moms, multiethnic, in urban USA population.
By chance, do authors know in these mother intending to breastfeed, is it known how long they intended to breastfeed before the interventions or when they were asked while in hospital?
For these 149 mothers, while in hospital, was donor breastmilk from a certified breastmilk bank an offer, or is donor breastmilk routinely used only in the setting of the babies of lower gestational age and in the neonatal intensive care unit?
Clarify if greater than 80 % breastfeeding is attempts, volume or both, especially if the babies received donor bank breastmilk.
Comment on breastfeeding in the first hour: baby -Mom had a doula or staff, baby in sniffing position, any instances during these feeds of unsafe sleep noted as the assumption there was not sudden postnatal collapse noted in the infants.
The conclusions are supported and important. Good discussion, and I read every refrence, good selection.
Author Response
Reviewer 3
Thank you for your comments. Our responses are written below in bold italics.
In physics and chemistry humaniity thought and studied what might be the ultimate building blocks of matter, and in biology, the fundamental unit of plant and animal life, the cell. With the knowledge in Medicine and biology that breastfeeding saes maternal and pediatric lives annually, the emergence of steps, 10 baby friendly not surprisingly emerged. The effectiveness of the steps, all of them, or a subset, would naturally emerge as its own area of study. The first nine Baby Friendly steps seem more valuable in the Congo based on a Melinda and Bill Gates supported study, for example. This study examines 149 post partum Moms, multiethnic, in urban USA population.
By chance, do authors know in these mother intending to breastfeed, is it known how long they intended to breastfeed before the interventions or when they were asked while in hospital?
Mothers were asked on hospital day 1 on their intention to breastfeed. We do not have data on how long they intended to breastfeed.
For these 149 mothers, while in hospital, was donor breastmilk from a certified breastmilk bank an offer, or is donor breastmilk routinely used only in the setting of the babies of lower gestational age and in the neonatal intensive care unit?
Donor breast milk is only offered to preterm infants of 34 weeks gestational age or less in the neonatal intensive care unit. Donor milk was not offered to any infants in our study of healthy term infants.
Clarify if greater than 80 % breastfeeding is attempts, volume or both, especially if the babies received donor bank breastmilk.
The babies did not receive donor breast milk so only feedings of breast milk at the breast were included.
Comment on breastfeeding in the first hour: baby -Mom had a doula or staff, baby in sniffing position, any instances during these feeds of unsafe sleep noted as the assumption there was not sudden postnatal collapse noted in the infants.
There were no recorded instances of unsafe sleep practices in the infants and no instances of postnatal collapse or illness.
The conclusions are supported and important. Good discussion, and I read every refrence, good selection.
Reviewer 4 Report
Congratulation! The article is of interest, because there are not many of this kind and they did not follow an appropriate study design which can be reproducible. A conclusion would be useful at the end.
Author Response
Reviewer 4
Thank you for your comments.
Congratulation! The article is of interest, because there are not many of this kind and they did not follow an appropriate study design which can be reproducible. A conclusion would be useful at the end.
A conclusion was added.
"Breastfeeding in the first hour and rooming-in were associated with increased high breastfeeding intensity. The findings of the present study support these Baby Friendly Hospital Interventions and should be incorporated into practice."
Round 2
Reviewer 1 Report
None.
Author Response
Thank you. There were no new comments.
Reviewer 2 Report
Thank you for the clarification on the suggestions and questions posed in the first round. This information is helpful and would be beneficial for readers. It would improve the manuscript to include many of the authors’ explanations in the main text, such as definitions, justifications, and which findings were significant and which were null. This information improves the validity of the findings and will increase the quality of the study in terms of rigor and reproducibility. I recommend adding to the manuscript text:
- That translation correction by consensus was completed by bilingual healthcare professionals with experience in lactation
- How participants were asked about postpartum breastfeeding interventions (line 88). The wording should be specific so that the reader knows what was being measured.
- The information pertaining to why 1st hour BF and rooming-in were part of the 1-month follow-up interview questions, and what happened when there was a discrepancy between the baseline and 1-month responses.
- The definitions of the confounding variables in the main text
- The justification for focusing on high-intensity BF vs duration – the explanation in the response was excellent and would be great information for readers
- No differences in BF intensity among Hispanic and non-Hispanic mothers
- That natality differences in BF intensity were tested, and why natality was not included in the final models
Author Response
Reviewer 2
Thank you for the clarification on the suggestions and questions posed in the first round. This information is helpful and would be beneficial for readers. It would improve the manuscript to include many of the authors’ explanations in the main text, such as definitions, justifications, and which findings were significant and which were null. This information improves the validity of the findings and will increase the quality of the study in terms of rigor and reproducibility. I recommend adding to the manuscript text:
Thank you for your comments. Our responses are written below in bold italics.
- That translation correction by consensus was completed by bilingual healthcare professionals with experience in lactation
This was added in line 89. “Any discrepancies in meaning were corrected by consensus on meaning and intent by bilingual healthcare professionals with expertise in lactation.”
- How participants were asked about postpartum breastfeeding interventions (line 88). The wording should be specific so that the reader knows what was being measured.
The following specific language is written in lines 94-98: “BF in the 1st hour of life was assessed by the question “When did you first put the baby to breast?” Rooming-in was optional and was assessed by asking the question “How often does the baby room-in with you?” followed by the choices: "all of the time", "some of the time", or "not at all." For this study rooming-in was defined as “all of the time.””
- The information pertaining to why 1st hour BF and rooming-in were part of the 1-month follow-up interview questions, and what happened when there was a discrepancy between the baseline and 1-month responses.
This was added in line 91-94. “ These questions were repeated at the one-month interview as the initial interview could have taken place early in the hospital stay and, therefore, may not have included what transpired for the rest of the hospital stay. If there was a discrepancy the one-month response was used.”
- The definitions of the confounding variables in the main text
This was added in line 112-115. “Maternal age was defined as age<25 vs 25 or above, marital status was defined as legally married at the time of delivery, overweight/obesity as BMI 25 or above, gravida as previous live birth, delivery type as NSVD vs. C-section or assisted vaginal delivery.”
- The justification for focusing on high-intensity BF vs duration – the explanation in the response was excellent and would be great information for readers
Thank you. This has been added in the introduction lines 59-61. “ In addition, measuring high breastfeeding intensity (whether 80% of all feeds are breast milk feeds), has not routinely been evaluated. The outcome measure of high breastfeeding intensity may be better at evaluating small differences in breastfeeding, especially in populations such as ours with a history of mixed feeding.”
- No differences in BF intensity among Hispanic and non-Hispanic mothers
Added in line 148-150. “ There were no differences between Hispanic and non-Hispanic mothers in high breast-feeding intensity in the hospital, 1 month and 3 months.”
- That natality differences in BF intensity were tested, and why natality was not included in the final models
Added in line 159-162. “Both nationality and having breastfed in the past showed a borderline difference on high BF intensity in the hospital (p=.062 and p=.057) and no difference at 1 and 3 months. As the results were similar, we only included breastfed in the past in the analysis.”